# Concurrent Validity and Test–Retest Reliability of the 3-Meter Backward Walk Test in Stroke Survivors

**DOI:** 10.3390/healthcare11233020

**Published:** 2023-11-22

**Authors:** Ali Kapan, Milos Ristic, Lin Yang, Gottfried Kranz, Thomas Waldhör

**Affiliations:** 1Center for Public Health, Department of Social and Preventive Medicine, Medical University of Vienna, 1090 Vienna, Austria; 2Neurological Rehabilitation Center Rosenhügel, 1130 Vienna, Austria; 3Department of Cancer Epidemiology and Prevention Research, Alberta Health Services, Calgary, AB T2S 3C3, Canada; 4Departments of Oncology and Community Health Sciences, University of Calgary, Calgary, AB T2S 3C3, Canada; 5Center for Public Health, Department of Epidemiology, Medical University of Vienna, 10090 Vienna, Austria

**Keywords:** 3-Meter Backward Walk Test, stroke, psychometric properties, concurrent validity, reliability

## Abstract

In the context of evaluating physical function in individuals with stroke, the 3-Meter Backward Walk Test (3MBWT) emerges as a potential tool of interest. The purpose of this study was to assess the test–retest reliability and concurrent validity of the 3MBWT and its correlation with falling incidents. Conducted in a neurological rehabilitation center, 35 ambulatory individuals with stroke were enrolled within a month post-stroke onset. These participants, with a Functional Ambulation Category score of ≥4, underwent the 3MBWT, Functional Gait Assessment (FGA), 10-Meter Walk Test (10MWT), and 6-Minute Walk Test (6MWT) under the supervision of different physiotherapists. The results indicate that the 3MBWT demonstrated high reliability, with an Intraclass Correlation Coefficient of 0.97 (95% CI: 0.95, 0.98). It also showed significant concurrent validity with other established walking tests like the 6MWT (r = −0.78) and 10MWT (r = 0.71), with a moderate correlation with the FGA (r = −0.54). No marked differences in test outcomes were observed between participants based on their fall history. Conclusively, the 3MBWT proves to be highly reliable and agrees well with existing walking function assessments for stroke patients, suggesting its potential as a time-efficient alternative.

## 1. Introduction

Balance impairment, with an estimated prevalence of 83% in stroke patients, can elevate fall risks and hinder daily activities [1]. Such imbalances after a stroke arise from factors like muscle weakness, abnormal muscle tone, sensory deficits, reduced attention, and vision and spatial awareness abnormalities [2,3]. Studies have found that stroke patients who possess the ability to stand frequently exhibit delayed and disrupted equilibrium reactions, exaggerated postural sway in both sagittal and frontal planes, and reduced weight bearing on the affected limb, all of which heighten their risk of falling [4]. Studies highlight that within the first six months post-discharge, between 36% and 73% of stroke survivors experience falls [5,6]. Furthermore, one year post-stroke, the incidence of falls persists at a notable level, with 40% to 58% of these individuals still experiencing falls [7,8]. These incidents not only raise the risk of injury but can lead to medical complications, extended rehabilitation, increased costs, and psychological distress, such as fear of further falls [9,10,11].

Consequently, reducing the risk of falls and their negative consequences is a crucial aspect of stroke rehabilitation. Given that balance impairments following a stroke are closely linked to the risk of falls, it is crucial to comprehensively assess the balance of affected individuals. Through such evaluations, at-risk patients can be identified, and appropriate rehabilitation measures can be planned. Previous studies have employed various tools to assess balance, including the Berg Balance Scale, Functional Reach Test, Timed Up and Go Test, Mini-BESTest, and Functional Gait Assessment (FGA) [12,13,14,15].

However, the majority of these assessment tools focus on forward walking and stepping, with the exception of the FGA, which includes a component for measuring backward walking. Backward walking has been found to demand higher levels of neuromuscular control, proprioception, and protective reflexes than forward walking, due to its greater complexity in motor planning and coordination [16]. Consequently, backward walking has been proposed as a more sensitive measure of balance and mobility [17,18]. Incorporating backward walking into balance assessment tools may offer a more comprehensive evaluation of balance in stroke patients. Such inclusion can capture deficits not evident in forward walking assessments alone, thus providing a holistic understanding of post-stroke balance impairments. Backward walking is a crucial component in assessing the physical function of stroke patients, yet it is often marginalized in gait assessments [19]. The 3-Meter Backward Walk Test (3MBWT) was developed to target this specific function. While its reliability has been assessed in earlier studies with stroke survivors [20,21], our research seeks to enhance its validity. We aim to achieve this by examining the concurrent validity of the 3MBWT in comparison with other established assessments, such as the FGA, 10-Meter Walk Test (10MWT), and 6-Minute Walk Test (6MWT). This comparative analysis is crucial to understand the applicability of the 3MBWT in the broader context of stroke rehabilitation, particularly for diverse patient groups with varied gait and balance impairments.

## 2. Methods

### 2.1. Study Design

The present study was conducted at the Neurological Rehabilitation Centre Rosenhügel Vienna between October 2022 and February 2023. All participants gave their written informed consent according to the Declaration of Helsinki, and the local ethics committee approved the study protocol.

The participants were provided with a thorough briefing on the procedures to ensure a clear understanding of the task requirements. Particularly for the 3MBWT, participants were allowed to acquaint themselves with the backward walking task by executing a few steps backward before the onset of the actual testing period. This familiarization aimed at evaluating the risk of falling during the backward walking task, as backward walking could potentially pose challenges for some individuals, thereby increasing the risk of falling. This was not designed as a practice session for the 3MBWT. Before conducting the 3MBWT, participants were allowed to rest while seated. The measurements were carried out under identical environmental conditions (e.g., consistent flooring, measurements in the morning, uniform lighting, and constant room size) to minimize possible influences from external factors. Due to organizational challenges at the rehabilitation center, it was not possible to assign the same two physiotherapists for the tests; therefore, five different physiotherapists were entrusted with conducting the tests. These conditions also precluded conducting an inter-rater reliability assessment. Given the possible variability arising from the involvement of five different physiotherapists, we opted for three repeated measurements of the 3MBWT according to our “test–retest” design (with appropriate recovery breaks of 2 to 5 min in between to restore energy levels and minimize the possible impact of fatigue on subsequent test performance). The test execution was timed with the same stopwatch, and participants were not informed of their recorded times. Upon completion of the three consecutive 3MBWT runs, the physiotherapists conducted further tests in the same session, including the 10MWT, the 6MWT, and the FGA. Unlike the 3MBWT, these assessments were conducted once and without a practice run, as forward walking does not require the same level of familiarity as walking backward.

### 2.2. Participants

The study enrolled patients of all genders who were at least 18 years old and diagnosed with hemorrhagic or ischemic stroke (within 1 month of onset) by a neurologist. Participants were required to have a Functional Ambulation Category (FAC) score of ≥4, indicating that they were able to walk without human assistance or devices [22], and to be oriented and cooperative by general condition. Patients who had a Mini-Mental State Examination (MMSE) score below 24, Alzheimer’s disease, similar dementia diseases, or were unable to follow instructions were excluded from the study.

### 2.3. Sample Size

The required sample size was determined using the Sample Size Calculator website https://wnarifin.github.io/ssc/ssalpha.html (accessed on 10 September 2023) [23]. Prior research by Abit et al. [20] and DeMark et al. [21] concerning the validity and reliability of the 3MBWT in stroke patients reported an ICC of 0.9. With such a high ICC, our required sample size would be only 16 participants. However, to adopt a more conservative approach and ensure the robustness of our findings, we based our calculations on a minimally acceptable ICC of 0.7. Using this more conservative estimate, with an alpha and beta set at 0.05 and 0.20, respectively, and accounting for a 10% anticipated dropout rate, we determined our study required a sample size of 33 ambulatory stroke survivors. Due to an administrative oversight, the total number of participants included in the study was 35, contrary to the initially stated 33. Data from all 35 participants were analyzed to ensure the robustness of the findings.

### 2.4. Measurement

#### 2.4.1. 3-Meter Backward Walk Test

For the 3MBWT, a distance of 3 m is measured and marked with black tape on the floor. Participants stand at the starting point with their heels aligned with the tape. Upon the command “walk”, they walk backward as quickly as possible, aiming to reach the 3-meter mark as accurately as possible before stopping. The elapsed time in seconds is recorded for each of the three trials, and participants are allowed to turn around during the test if they wish to do so. However, running during the test is not allowed to ensure safety and prevent falls. Throughout the entire test, the evaluator walks behind the participants to ensure their safety and prevent potential falls. A shorter time spent in competing the 3MBWT indicates a better performance on the test [20,21].

#### 2.4.2. Functional Gait Assessment

The FGA is a performance-based measure that assesses the ability to perform various walking tasks in a manner that reflects the complexity of everyday life activities. It is composed of 10 tasks that require different levels of balance control, including walking with changing speed, walking backward, walking while looking up and down, walking and turning, walking over obstacles, and walking on uneven surfaces. Each task is scored on a 4-point ordinal scale (0 to 3), with higher scores indicating a better performance. The maximum score is 30 points, with each task worth up to 3 points. The FGA takes about 15–20 min to administer and has been found to be a reliable and valid measure of gait function in different patient populations, including stroke patients. It is considered to be a comprehensive and sensitive tool for assessing gait function and has been used in clinical and research settings to evaluate interventions and track changes in gait function over time [15,24].

#### 2.4.3. 10-Meter Walking Test

The 10MWT was conducted by identifying a 14-meter path of flat, hard-unobstructed surface in the laboratory room. The first and last two meters were used as acceleration and deceleration zones to allow for a uniform walking pace. Four cones marked the beginning and end of the path as well as the start and end of the acceleration and deceleration zones. Each participant received a walking demonstration before the start of the test, but they did not have a practice walk themselves. The following standardized instructions were given: “This is our walking corridor. I want you to walk to the other end of the trail at a comfortable speed, as if you were walking down the street. Walk past the other end of the tape before stopping”. With the instructions of “Ready, steady, go”, the participant began to walk. Participants were allowed to use their usual walking aid if needed and started behind the starting line with their toes touching the line. The 10MW aims to assess forward gait speed, which can predict morbidity and mortality as well as functional ability [25,26].

#### 2.4.4. 6-Minute Walk Test

The 6MWT is typically conducted in a long, straight corridor that is at least 30 m long. However, in our rehabilitation facility, a corridor of 150 m in length was used for the 6MWT, where patients could complete multiple laps as needed. During the test, the participant was instructed to walk as many laps as possible along the corridor for 6 min. The participant was encouraged to walk at his or her own pace and to rest if necessary, but the timer continued to run throughout the entire 6 min. The distance covered in meters during the 6 min was recorded and used as a measure of functional capacity and endurance. The 6MWT has been shown to be a reliable and valid measure of functional performance in a variety of patient populations [26].

#### 2.4.5. Functional Ambulation Categories

The FAC is a functional walking test that assesses ambulation ability. It utilizes a 6-point scale to determine the level of human support a patient requires while walking, irrespective of their use of personal assistive devices. It is important to note that the FAC does not measure endurance, as the patient is only required to walk a distance of approximately 3 m. Although the FAC is commonly employed with stroke patients, it can be utilized with other populations as well [22].

Demographic data (age, height, and weight), affected and dominant side, and type and duration of stroke were recorded. Based on the recorded number of falls in the past two months, participants were categorized as “Fallers” if they had experienced one or more falls, and “Non-Fallers” if they had not experienced any falls. Furthermore, the wellbeing and gait safety of individuals were assessed using a single-item question rated on a scale ranging from 1 (very poor) to 10 (very good).

### 2.5. Statistical Analysis

Descriptive statistics were employed to depict the characteristics of the sample population. Continuous variables were reported as means, standard deviations (SDs), and medians with corresponding minimum and maximum values (min–max). Categorical variables were presented as frequencies and percentages. The Intraclass Correlation Coefficient (ICC) and its 95% Confidence Interval (CI) were calculated for the three repetitions of the 3MBWT using a random-effects model, with patient ID as the random effect. The ICC was calculated as the ratio var(random effect)/(var(random effect) + var(residual)). Bland–Altman plots were constructed to examine the limits of agreement and to assess any systematic bias between the two sessions [27].

The concurrent validity was assessed by analyzing the correlations between the 3MBWT and the FGA, 10MWT, and 6MWT measures using either the Pearson correlation coefficient (r) or the Spearman correlation coefficient (rho), as appropriate. The resulting correlation coefficients were classified as poor (0.00–0.25), fair (0.26–0.50), moderate (0.51–0.75), or strong (0.76–1.00), according to established criteria [28].

To investigate the potential for learning or fatigue effects across the three repetitions of the 3MBWT, we conducted an analysis of variance (ANOVA). This allowed us to assess the impact of repeated measures on the test performances. Further, an unpaired Wilcoxon test was used to compare the location parameter between patients with a history of falls and those without. The significance level was set at *p* = 0.05, and *p*-values were not adjusted for multiple testing. All statistical calculations were performed using SAS version 9.4 (SAS Institute Inc., Cary, NC, USA).

## 3. Results

Table 1 provides a detailed summary of the socio-demographic characteristics of the study participants, consisting of a total of 35 individuals, with females constituting 42.9% of the sample. The mean age of participants was 64.9 years (SD = 13.9), and the average BMI was recorded at 26.3 kg/m^2^ (SD = 4.8). When it comes to the specifics of stroke, ischemic stroke was the most common type, affecting 94.3% of participants, while 5.7% had suffered a hemorrhagic stroke. Participants were predominantly right-side-dominant (97.1%) with the affliction of stroke distributed almost equally between the left (51.4%) and the right (48.6%) sides.

The results from the 3MBWT in Table 2 showed a slight decrease in mean times across the three rounds, from an initial 11.6 s to 10.9 s in the last round. In addition, the FGA revealed a median score of 24, while participants achieved a median completion time of 9 s on the 10MWT. For the 6MWT, participants covered an average distance of 367.3 m. The conducted ANOVA revealed no significant differences between the three consecutive runs of the mean 3MBWT (*p* = 0.768), indicating that neither learning nor fatigue effects played a substantial role in the participants’ test performance.

The ICC and 95% CI for the three repetitions were estimated at 0.97 (0.95, 0.98), indicating very small variation within repeated measurements but large variation between patients. The Bland–Altman analysis of the 3MBWT across three attempts highlighted consistent agreement patterns. Moderate agreement was noted between the first and second attempts, with average differences and limits of agreement (see Figure 1a–c red dashed lines) within acceptable ranges. This trend was similar in comparisons between the first and third and the second and third attempts. However, an increased spread of differences was observed at higher mean values, particularly above 20 s, suggesting the possibility of systematic or proportional errors for longer test durations.

In the correlation analysis (Figure 2), the 3MBWT showed a negative correlation with the 6MWT (r = −0.78, 95% CI −0.88 to −0.60), indicating that individuals who took longer to complete the 3MBWT tended to cover shorter distances on the 6MWT. A positive correlation was found with the 10MWT (r = 0.71, 95% CI 0.49 to 0.84), with longer 3MBWT completion times associated with increased times in the 10MWT. Additionally, a negative correlation with the FGA (r = −0.54, 95% CI −0.74 to −0.26) suggests that longer times on the 3MBWT could be linked to lower FGA scores. Conversely, FGA scores were positively correlated with distances covered in the 6MWT (r = 0.67) and negatively correlated with times in the 10MWT (r = −0.77). These correlations suggest that higher FGA scores are associated with longer distances in the 6MWT and shorter times in the 10MWT. Furthermore, an inverse relationship was identified between the 10MWT times and the 6MWT distances, with a correlation coefficient of −0.86, indicating that longer times in the 10MWT are associated with shorter distances in the 6MWT.

Comparisons between patients with and without history of falls did not reveal significant differences in the 3MBWT (*p* = 0.136), FGA score (*p* = 0.532), 10 MWT (*p* = 0.173), or 6MWT (*p* = 0.108).

## 4. Discussion

In our study, the ICC for the three repetitions of the 3MBWT was estimated at 0.97 (95% CI: 0.95, 0.98), demonstrating minimal variation within repeated measurements, which was remarkably consistent across five different physiotherapists. This high reliability, achieved even with multiple therapists involved, underscores the robustness of the 3MBWT in diverse clinical settings. Despite potential concerns about variability due to the absence of an inter-rater reliability assessment, our findings indicate strong intra-rater reliability. Our findings align with those of DeMark et al. [21], who reported a large intra-rater correlation of 0.96 in both subacute and chronic stroke patient groups, suggesting a similar level of reliability. Additionally, Abit et al. [20], reported excellent internal consistency with a Cronbach’s alpha of 0.97, further supporting the 3MBWT’s efficacy. Our analysis of the Bland–Altman plots revealed a range of around ±5 s in the limits of agreement, albeit with some outliers. These outliers, which may represent individuals with more pronounced disabilities or those at a later stage in life, serve to highlight the range of variability naturally present in clinical evaluations. By incorporating data from all patients in our study, we aimed to offer a comprehensive and realistic depiction of the 3MBWT’s clinical utility. This approach allowed us to cover a wide array of patient experiences, underscoring the importance of tailoring interpretations of test results to each individual’s unique circumstances.

The ANOVA results indicate the absence of significant differences between the repeated measurements, suggesting that neither a learning effect, which would mean a reduction in time over three trials, nor fatigue, which would imply longer times, significantly affected performance. This means the performance in three consecutive 3MBWT measurements within a single session remained constant, alleviating concerns regarding learning or fatigue effects. These findings underscore the potential of the 3MBWT as a reliable assessment tool in clinical settings, particularly for tailoring therapeutic interventions to individual patients.

The assessment of mobility and balance in stroke patients is crucial in the clinical setting for establishing accurate diagnoses, planning individual treatment methods, and evaluating the effectiveness of rehabilitation [29]. In our research, selecting assessments that encapsulate a wide range of gait parameters was crucial to achieving a comprehensive evaluation and to enhancing the robustness of our findings. The FGA, 10MWT, and 6MWT were chosen as they spotlight different facets of gait functions, enabling a more nuanced exploration of gait parameters [15,29]. Our results demonstrate similarly strong correlations between the 3MBWT and the 6MWT (r = −0.78), as well as between the 3MBWT and the 10MWT (r = 0.71). These similar strengths in correlations suggest that the duration of the 3MBWT is comparably associated with both the distance covered in the 6MWT and the time required in the 10MWT. This may indicate that factors influencing performance in the 3MBWT, such as balance and coordination, are equally relevant for performance in both time-based and distance-based walking tests. While the 6MWT primarily assesses endurance and functional mobility and the 10MWT focuses on walking speed, the similar correlation strengths imply that both aspects—endurance and speed—are closely linked when assessing mobility in stroke survivors. This emphasizes the need to consider both aspects in rehabilitation.

Furthermore, the positive correlation between FGA scores and distances covered in the 6MWT (r = 0.676), along with the negative correlation with times in the 10MWT (r = −0.776), suggests that higher FGA scores correspond to a better walking performance. These findings are consistent with the correlations observed between the 3MBWT and the other two tests (6MWT and 10MWT), where longer durations in the 3MBWT indicate a worse walking performance. The similarity of these correlations underscores the complex nature of gait disorders in stroke patients and highlights the need to consider multiple aspects of mobility in clinical assessments.

A strong correlation between the performances in the 6MWT and the 10MWT was anticipated, as previously observed in studies involving stroke patients. These results suggest that both the 6MWT and the 10MWT measure similar aspects of walking ability, being largely influenced by neural impairments [25,26]. Additionally, the correlation between the 3MBWT and the FGA was found to be moderate. This suggests that while there is an association, the 3MBWT and the FGA may measure different aspects of gait and balance abilities, indicating that they might not be interchangeable assessments in a clinical setting. The 3MBWT focuses on the ability to walk backward, requiring balance, coordination, and proprioceptive abilities [16]. The FGA, on the other hand, measures a broader spectrum of gait and balance abilities, including the ability to perform head and neck movements while walking. This requires additional skills, such as integrating vestibular information and maneuvering the body to maintain stability during these added tasks [24]. Consequently, individuals who are capable of efficiently walking backward (and performing well in the 3MBWT) might encounter difficulties in performing complex tasks during the FGA, resulting in a moderate correlation between the two tests.

The results suggest that while the various walking tests measure different aspects of gait ability, they are strongly interconnected. This demonstrates the complexity of gait disorders in stroke patients and the necessity of assessing multiple parameters in order to obtain a comprehensive picture of mobility.

Our study aimed to form a relatively homogeneous group of stroke patients with an FAC score of 4 or 5. This approach was intended to maintain consistency in terms of reliability and validity throughout our research. This objective was reflected in the low number of observed falls, with approximately 25% of participants reporting a fall within the last two months, a figure lower than the 36% to 73% reported in the stroke patient literature [5,6]. As a result, this limited occurrence could have restricted the statistical power of the assessments conducted on the scores of the 3MBWT and FGA between stroke patients with and without a history of falls, resulting in non-significant *p*-values. These findings suggest that the low incidence of falls in our study group may have influenced the outcomes, limiting our ability to detect potential differences. It is crucial to consider that there may be unmeasured factors, such as muscle strength, depression, medication effects, or fear of falling, which could have a substantial impact on the risk of falls and should be considered in future investigations. While we attempted to account for some potential confounding variables, our single-item approach may not fully capture the complexity of fall risk. However, it is important to note that falls occurring in a backward direction have been reported in previous studies [30]. Slowing of backward walking speed has also been observed in older individuals and has been associated with an increased risk of falls [16,31]. These findings suggest that the 3MBWT may indeed be a valuable tool for assessing fall risk in stroke patients. Additionally, the lack of association between the 3MBWT implementation time and fall frequency extends to the broader results of the FGA, highlighting the complexity of fall risk in stroke patients. Notably, the FGA has demonstrated the ability to predict falls within a 6-month period in a study involving community-dwelling older adults, exhibiting high sensitivity (100%) and specificity (72%) [32]. While the association between FGA scores and fall risk in stroke patients has not been consistently confirmed, a moderate negative correlation has been found, suggesting that the FGA may have some predictive value in predicting fall risk in stroke patients [33]. Therefore, conducting a more comprehensive assessment is crucial to gaining a better understanding of fall risk in stroke patients.

This study has several limitations that should be acknowledged. Firstly, the sample size was small, which limited our ability to detect differences between patients with and without a history of falls. Additionally, the study focused on a specific subset of stroke patients with a minimum FAC score of ≥4. While this homogeneity is advantageous for reliability and validity studies, it may restrict the external validity and generalizability to stroke patients with more severe impairments. Another limitation of this study involves the lack of an inter-rater reliability assessment for 3MBWT. Although our study showed robust intra-rater reliability, with an ICC of 0.97 (95% CI: 0.95, 0.98) for the three repetitions of the 3MBWT, it is noteworthy that, owing to the study’s design, we were unable to directly evaluate inter-rater agreement. Despite efforts to mitigate fatigue effects by providing adequate rest periods, the lack of randomization in the sequence of different tests conducted within the same day remains a limitation of this study. Future investigations should consider randomizing the order of various tests to further diminish the influence of learning and fatigue effects and to improve the study’s internal validity.

## 5. Conclusions

The study confirms the 3MBWT to be a reliable and valid instrument for assessing mobility in stroke survivors, exhibiting high test–retest reliability. It shows a strong concurrent validity with the 10MWT and 6MWT and a moderate correlation with the FGA, indicating its effectiveness in capturing distinct yet interrelated aspects of gait and balance. These results endorse the 3MBWT’s integration into stroke rehabilitation assessments, providing a comprehensive evaluation of mobility challenges faced by stroke patients and suggesting its potential as a time-efficient alternative.

## Figures and Tables

**Figure 1 healthcare-11-03020-f001:**
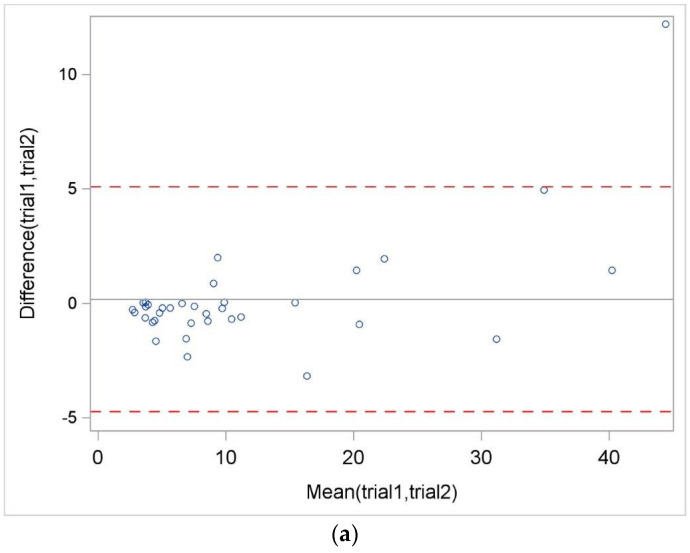
(**a**) Bland–Altman plots between trial 1 and 2. (**b**) Bland–Altman plots between trial 1 and 3. (**c**) Bland–Altman plots between trial 2 and 3.

**Figure 2 healthcare-11-03020-f002:**
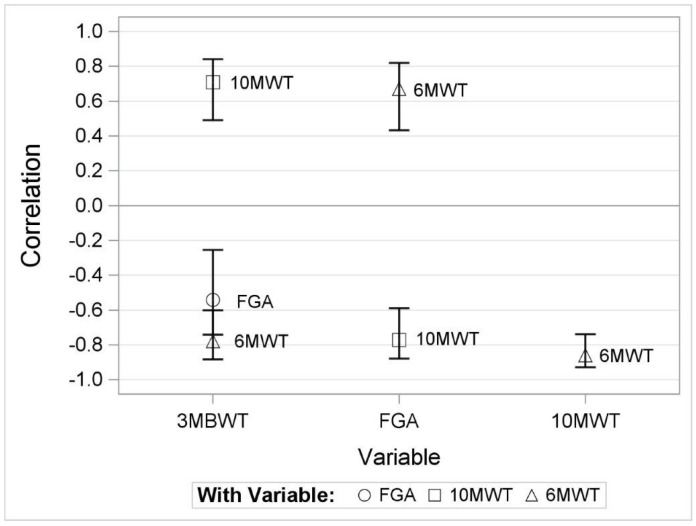
Correlation coefficients and corresponding 95% CI.

**Table 1 healthcare-11-03020-t001:** Demographic and medical information.

Variables	All Participants (n = 35)	Non-Fallers (n= 26)	Fallers (n = 9)
Female, n (%)	15 (42.9)	8 (30.8)	7 (77.8)
Age (years), mean (SD)	64.9 (13.9)	62.0 (14.6)	73.3 (7.6)
BMI (kg/m^2^), mean (SD)	26.3 (4.8)	26.1 (4.6)	26.9 (5.5)
Stroke duration (month), median (min–max)	3 (1–97)	3 (1–97)	2.67 (1–10)
Stroke type, n (%)			
Hemorrhagic	2 (5.7)	2 (7.7)	0 (0)
Ischemic	33 (94.3)	24 (92.3)	9 (100)
Subacute phase	27 (77.1)	19 (73.1)	8 (88.9)
Chronic phase	8 (22.9)	7 (26.9)	1 (11.1)
Dominant side, n (%)			
Right	34 (97.1)	25 (96.2)	9 (100)
Affected side n (%)			
Right	17 (48.6)	14 (53.8)	3 (33.3)

Note: SD: standard deviation, BMI: Body Mass Index.

**Table 2 healthcare-11-03020-t002:** Functional assessments and test results.

Variables	All Participants (n = 35)	Non-Fallers (n = 26)	Fallers (n = 9)
FAC Score n (%)			
FAC 4	10 (28.6)	6 (23.1)	4 (44.4)
FAC 5	25 (71.4)	20 (76.9)	5 (55.6)
3MBWT (seconds)—1 round, mean (SD)	11.6 (10.1)	10.3 (9.3)	15.5 (12.5)
3MBWT (seconds)—2 round, mean (SD)	11.8 (11.7)	10.3 (10.7)	16.2 (14.1)
3MBWT (seconds)—3 round, mean (SD)	10.9 (10.3)	9.3 (8.9)	15.9 (12.8)
FGA Score, median (min–max)	24 (4–30)	24 (10–30)	20 (4–28)
10-Meter Walk Test (seconds), median (min–max)	9 (5.9–27.0)	9 (5.9–19)	12.5 (6–27)
6-Minute Walk Test (meters), mean (SD)	367.3 (132.5)	387.0 (122.5)	310.5 (151.2)
Single-Question Wellbeing (Scale 1–10) median (min–max)	8 (1–10)	8 (5–10)	6 (1–10)
Single-Question Gait safety (Scale 1–10) median (min–max)	7 (1–10)	8 (3–10)	6 (1–8)

Note: 3MBWT: 3 Meter Backward Walk Test; FGA: Functional Gait Assessment, FAC: Functional Ambulation Categories.

## Data Availability

The data presented in this study are available on request from the corresponding author.

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
