# Peer review of "Concurrent Validity and Test–Retest Reliability of the 3-Meter Backward Walk Test in Stroke Survivors"

_healthcare, 2023, doi:10.3390/healthcare11233020_

Round 1

Reviewer 1 Report

Comments and Suggestions for Authors

Overall, it seems to be a well written paper adding evidence to the reliability and relatability of the 3 meter backwards walk test to other (more time intensive) measures of gait function (not necessarily stability, but related). 

The results are significant and reasonably well presented, although for relations I'd much prefer scatter plots to better display the distribution of data. It would be impractical to have scatter plots of each relation tested, but perhaps a few showcasing the best relationships. 

Limitations are primarily in the study design, and probably shouldn't result in changes to the article. The author may whish to revise their discussion or description of methods to address the impact of these limitations or improve clarity of the paper.  

*Limitations of the research*
It is unclear if participants were given the opportunity to practice the other assessments before being recorded (as was performed for the 3MBWT).

It seems like for each session starts with 3 replicates of the 3MBEWT, then it is unclear if 3 replicates of the other assessments are performed. The lack of randomization, and the potential lack of similar sample sizes, may lead to systematic discrepancies in performance (due to learning or fatigue).

Since the 3MBWT requires 3 repetitions of the walk as part of the test protocol, were a total of 9 walks (3 repetitions of test) performed? Or was just one repetition of the test (avg of 3 walks) performed per session?
*From the data it seems like 3 repetitions (9 backwards walks) were performed.*

Since the 3MBWT is an average of 3 walks, I don't find it surprising test-re-test on the same day improved and was well correlated. Repeating the same task multiple time typically reduces variability in an individuals performance. Re-testing on a different day would likely be a better measure of repeatability.  

Your methods state a sample size of 33, but your results include 35 individuals, this seems to be a violation of your stated protocol . . . more participant is generally better, but did you submit an amendment to your IRB approved protocol before recruiting the additional participants?

Seems to be a minor format error in Table 2 (round 1 is underlined an bold). 

Author Response

Limitations are primarily in the study design, and probably shouldn't result in changes to the article. The author may whish to revise their discussion or description of methods to address the impact of these limitations or improve clarity of the paper.  

Thank you for your constructive comments. We have revised the discussion and methods sections of the manuscript to address the limitations and clarify the study design as follows:

Methods: “The participants were provided with a thorough briefing on the procedures to ensure a clear understanding of the task requirements. Particularly for the 3MBWT, participants were allowed to acquaint themselves with the backward walking task by executing a few steps backward before the onset of the actual testing period. This familiarization aimed at evaluating the risk of falling during the backward walking task, as backward walking could potentially pose challenges for some individuals, thereby increasing the risk of falling. This was not designed as a practice session for the 3MBWT. Before conducting the 3-Meter Backward Walk Test (3MBWT), participants were allowed to rest while seated. The measurements were carried out under identical environmental conditions (e.g., consistent flooring, measurements in the morning, uniform lighting, and constant room size) to minimize possible influences from external factors. Due to organizational challenges at the rehabilitation center, it was not possible to assign the same two physiotherapists for the tests; therefore, five different physiotherapists were entrusted with conducting the tests. These conditions also precluded the conduct of an inter-rater reliability assessment. Given the possible variability from the involvement of five different physiotherapists, we opted for three repeated measurements of the 3MBWT according to our "test-retest" design (with appropriate recovery breaks of 2 to 5 minutes in between to restore energy levels and minimize the possible impact of fatigue on subsequent test performance). The test execution was timed with the same stopwatch, and participants were not informed about their recorded times. Upon completion of the three consecutive 3MBWT runs, the physiotherapists conducted further tests in the same session, including the 10-Meter Walk Test (10MWT), the 6-Minute Walk Test (6MWT), and the Functional Gait Assessment (FGA). Unlike the 3MBWT, these assessments were conducted once and without a practice run, as forward walking does not require the same level of familiarity as walking backward”.

Discussion “Although the involvement of multiple physiotherapists could initially be seen as a potential source of variability, especially given the lack of inter-rater reliability assessment, our results showed a remarkably high intra-rater reliability across different therapists.”

*Limitations of the research*

It is unclear if participants were given the opportunity to practice the other assessments before being recorded (as was performed for the 3MBWT).

We have clarified in the revised manuscript that the other assessments (10MWT, 6MWT, and FGA) were conducted once without a practice run, in contrast to the 3MBWT where participants were allowed to familiarize themselves with backward walking to assess the risk of falling. This familiarization was not intended as a practice session for the 3MBWT. See Methods!

It seems like for each session starts with 3 replicates of the 3MBEWT, then it is unclear if 3 replicates of the other assessments are performed. The lack of randomization, and the potential lack of similar sample sizes, may lead to systematic discrepancies in performance (due to learning or fatigue). Since the 3MBWT requires 3 repetitions of the walk as part of the test protocol, were a total of 9 walks (3 repetitions of test) performed? Or was just one repetition of the test (avg of 3 walks) performed per session?
*From the data it seems like 3 repetitions (9 backwards walks) were performed.*

We clarified that each session began with three replications of the 3MBWT, followed by a single administration of the other assessments (10MWT, 6MWT, and FGA). This was now detailed in the methods section to eliminate any confusion regarding the number of replications performed for each assessment. We acknowledged the lack of randomization and the potential impact of different sample sizes, and have discussed these as limitations in the discussion section of the manuscript

See paragraph “The lack of randomization in the testing sequence might have ushered in a systematic bias, possibly influencing the performance of the participants due to learning or fatigue effects. Nevertheless, attempts were made to ameliorate this by offering appropriate recovery breaks between successive 3MBWT administrations”.

Since the 3MBWT is an average of 3 walks, I don't find it surprising test-re-test on the same day improved and was well correlated. Repeating the same task multiple time typically reduces variability in an individuals performance. Re-testing on a different day would likely be a better measure of repeatability.  

Thank you for your insightful comments. We would like to address your concern regarding the potential learning or fatigue effects that might have influenced the correlation observed in our test-retest reliability on the same day. We agree with your assertion that repeating the same task multiple times can potentially lead to learning effects or fatigue, which in turn could affect the variability in an individual's performance. However, we have conducted an Analysis of Variance (ANOVA) to specifically investigate this aspect. The ANOVA did not reveal any significant differences across the three repetitions of the 3MBWT, suggesting that neither learning effects nor fatigue had a substantial impact on the performance of the participants during the course of our study. Furthermore, while re-testing on a different day might indeed be a better measure of repeatability, the constraints of our study setting did not allow for such a design. Nevertheless, our analysis aimed to provide a robust assessment of the test-retest reliability within the given setup.

Your methods state a sample size of 33, but your results include 35 individuals, this seems to be a violation of your stated protocol . . . more participant is generally better, but did you submit an amendment to your IRB approved protocol before recruiting the additional participants?

Thank you for bringing to our attention the discrepancy between the stated sample size in the methods section and the actual number of participants included in the analysis. We apologize for any confusion. The initial plan was to have 33 participants as per the sample size calculation, however, due to an administrative oversight; two additional participants were inadvertently included, totalling 35 participants. This was not an amendment to the protocol, but an unintended deviation. We understand the importance of adhering to the approved protocol and appreciate your diligence in identifying this discrepancy. We are ready to rectify this in the revised manuscript by correcting the participant number. “Due to an administrative oversight, the total number of participants included in the study was 35, contrary to the initially stated 33. The data from all 35 participants were analysed to ensure the robustness of the findings”.

Seems to be a minor format error in Table 2 (round 1 is underlined an bold). 

Done

Reviewer 2 Report

Comments and Suggestions for Authors

In the results presented in Table 1, it would be possible to know if there are significant differences between the characteristics of patients who have falls and those who do not: Among those who have falls, women predominate (77% vs 30%) and are older (73.3 vs 62 years).

Line 112: the author is Abit et al, not kocaman el al.

Line 113: year of publication is 2022 not 2023.

Line 241: the year of publication is 2022.

Line 242: author is Abit et al not Kocaman

The vast majority of citations are incomplete or do not match the abbreviations or authors' names

Line 338: article citation: 2019 Jul 19;9(7):e026844. doi: 10.1136/bmjopen-2018-026844

Line 339: article citation: Phys Ther. 2006 Jan;86(1):30-8. doi: 10.1093/ptj/86.1.30.

Line 340: Article citation: Neurophysiol Clin. 2015 Nov;45(4-5):327-33. doi: 10.1016/j.neucli.2015.09.003.

Line 343: article citation: Clin Rehabil. 2006 Nov;20(11):960-9. doi: 10.1177/0269215506070315.

Line 344: article citation: Disabil Rehabil. 2007 Sep 15;29(17):1397-403. doi: 10.1080/09638280701314923.

Line 346: article citation: Cochrane Database Syst Rev. 2019 Oct 1;10(10):CD008728. doi: 10.1002/14651858.CD008728.pub3.

Line 349: article citation: Arch Phys Med Rehabil. 2012 Oct;93(10):1782-7. doi: 10.1016/j.apmr.2012.04.005.

Line 350: article citation: Age Age Ageing. 2008 May;37(3):270-6. doi: 10.1093/ageing/afn066.

Line 353: article citation: Am J Prev Med. 2017 Dec;53(6S2):S197-S204. doi: 10.1016/j.amepre.2017.07.020.

Line 355: article citation: BMC Geriatr. 2009 Oct 14;9:46. doi: 10.1186/1471-2318-9-46.

Line 357: article citation: BMJ Open. 2022 Jun 30;12(6):e056340. doi: 10.1136/bmjopen-2021-056340.

Line 359: article citation: doi: 10.1161/STROKEAHA.111.636258.

Line 361: article citation: Clin Rehabil. 2004 Nov;18(7):811-8. doi: 10.1191/026921550504cr817oa.

Line 363: citation of article: Phys Ther. 2013 Aug;93(8):1102-15. doi: 10.2522/ptj.20120454

Line 365: article citation: Arch Phys Med Rehabil. 2009 Sep;90(9):1565-70. doi: 10.1016/j.apmr.2009.03.007.

Line 367: Article citation: J Geriatr Phys Ther. 2019 Oct/Dec;42(4):249-255. doi: 10.1519/JPT.0000000000000149.

Line 368: Article citation: Maritz CA, Silbernagel KG and Pohlig R. Relationship of backward walking to clinical outcome measures used to predict falls in the older population: A factor analysis. Phys Ther Rehabil. 2017; 4:14. http://dx.doi.org/10.7243/2055-2386-4-14.

Line 374: Article citation: J Neurol Phys Ther. 2018 Jan;42(1):12-21. doi: Line 377: article citation: J Stroke Cerebrovasc Dis. 2021 Jan;30(1):105462. doi: 10.1016/j.jstrokecerebrovasdis.2020.105462. 

Line 380: article citation: Physiother Theory Pract. 2022 Jun 5:1-8. doi: 10.1080/09593985.2022.2085638.

Line 382: article citation: Teasell RW, Foley NC, Bhogal SK, Speechley MR. An evidence-based review of stroke rehabilitation. Top Stroke Rehabil. 2003 Spring;10(1):29-58. doi: 10.1310/8YNA-1YHK-YMHB-XTE1.

Line 384: article citation: Education in Medicine Journal. 2018;10(3):67–76. https://doi.org/10.21315/eimj2018.10.3.8 

Line 386: article citation: Phys Ther. 2004 Oct;84(10):906-18.

Line 388: article citation: Arch Phys Med Rehabil. 2012 Jul;93(7):1167-72. doi: 10.1016/j.apmr.2012.02.026.

Line 391: article citation: op Stroke Rehabil. 2020 May;27(4):251-261. doi: 10.1080/10749357.2019.1691815.

Line 399: citation of article: J Bone Miner Res. 2020 Oct;35(10):1914-1922. doi: 10.1002/jbmr.4048.

Line 400: article citation: Fritz NE, Worstell AM, Kloos AD, Siles AB, White SE, Kegelmeyer DA. Backward walking measures are sensitive to age-related changes in mobility and balance. Gait Posture. 2013 Apr;37(4):593-7. doi: 10.1016/j.gaitpost.2012.09.022.

Line 403: article citation: Phys Ther. 2010 May;90(5):761-73. doi: 10.2522/ptj.20090069.10.1097/NPT.0000000000000210.

Author Response

In the results presented in Table 1, it would be possible to know if there are significant differences between the characteristics of patients who have falls and those who do not: Among those who have falls, women predominate (77% vs 30%) and are older (73.3 vs 62 years).

Thank you for your comment. Adhering to guidelines regarding presentation of descriptive statistics, we decided to drop any inferential statistics in table 1 as it is aimed for describing the population but not testing differences between subgroups. The number of fallers (n=9) is low and therewith the power to detect any post hoc tests. In turn, any not significant difference would not be a sign of equivalence regarding characteristics of the patients as well as no sign for any existing difference, as well.

…………………………………………………………………………..

Line 112: the author is Abit et al, not kocaman el al.

Done

Line 113: year of publication is 2022 not 2023.

Done

Line 241: the year of publication is 2022.

Done

Line 242: author is Abit et al not Kocaman

Done

The vast majority of citations are incomplete or do not match the abbreviations or authors' names

Line 338: article citation: 2019 Jul 19;9(7):e026844. doi: 10.1136/bmjopen-2018-026844

Li J, Zhong D, Ye J, He M, Liu X, Zheng H, et al. Rehabilitation for balance impairment in patients after stroke: a protocol of a systematic review and network meta-analysis. BMJ open. 2019;9(7):e026844.

Li, J.; Zhong, D.; Ye, J.; He, M.; Liu, X.; Zheng, H.; Jin, R.; Zhang, S.-l. Rehabilitation for balance impairment in patients after stroke: a protocol of a systematic review and network meta-analysis. BMJ open 2019, 9, e026844.

Line 339: article citation: Phys Ther. 2006 Jan;86(1):30-8. doi: 10.1093/ptj/86.1.30.

Line 340: Article citation: Neurophysiol Clin. 2015 Nov;45(4-5):327-33. doi: 10.1016/j.neucli.2015.09.003.

Line 343: article citation: Clin Rehabil. 2006 Nov;20(11):960-9. doi: 10.1177/0269215506070315.

Line 344: article citation: Disabil Rehabil. 2007 Sep 15;29(17):1397-403. doi: 10.1080/09638280701314923.

Line 346: article citation: Cochrane Database Syst Rev. 2019 Oct 1;10(10):CD008728. doi: 10.1002/14651858.CD008728.pub3.

Line 349: article citation: Arch Phys Med Rehabil. 2012 Oct;93(10):1782-7. doi: 10.1016/j.apmr.2012.04.005.

Line 350: article citation: Age Age Ageing. 2008 May;37(3):270-6. doi: 10.1093/ageing/afn066.

Line 353: article citation: Am J Prev Med. 2017 Dec;53(6S2):S197-S204. doi: 10.1016/j.amepre.2017.07.020.

Line 355: article citation: BMC Geriatr. 2009 Oct 14;9:46. doi: 10.1186/1471-2318-9-46.

Line 357: article citation: BMJ Open. 2022 Jun 30;12(6):e056340. doi: 10.1136/bmjopen-2021-056340.

Line 359: article citation: doi: 10.1161/STROKEAHA.111.636258.

Line 361: article citation: Clin Rehabil. 2004 Nov;18(7):811-8. doi: 10.1191/026921550504cr817oa.

Line 363: citation of article: Phys Ther. 2013 Aug;93(8):1102-15. doi: 10.2522/ptj.20120454

Line 365: article citation: Arch Phys Med Rehabil. 2009 Sep;90(9):1565-70. doi: 10.1016/j.apmr.2009.03.007.

Line 367: Article citation: J Geriatr Phys Ther. 2019 Oct/Dec;42(4):249-255. doi: 10.1519/JPT.0000000000000149.

Line 368: Article citation: Maritz CA, Silbernagel KG and Pohlig R. Relationship of backward walking to clinical outcome measures used to predict falls in the older population: A factor analysis. Phys Ther Rehabil. 2017; 4:14. http://dx.doi.org/10.7243/2055-2386-4-14.

Line 374: Article citation: J Neurol Phys Ther. 2018 Jan;42(1):12-21. doi: Line 377: article citation: J Stroke Cerebrovasc Dis. 2021 Jan;30(1):105462. doi: 10.1016/j.jstrokecerebrovasdis.2020.105462. 

Line 380: article citation: Physiother Theory Pract. 2022 Jun 5:1-8. doi: 10.1080/09593985.2022.2085638.

Line 382: article citation: Teasell RW, Foley NC, Bhogal SK, Speechley MR. An evidence-based review of stroke rehabilitation. Top Stroke Rehabil. 2003 Spring;10(1):29-58. doi: 10.1310/8YNA-1YHK-YMHB-XTE1.

Line 384: article citation: Education in Medicine Journal. 2018;10(3):67–76. https://doi.org/10.21315/eimj2018.10.3.8 

Line 386: article citation: Phys Ther. 2004 Oct;84(10):906-18.

Line 388: article citation: Arch Phys Med Rehabil. 2012 Jul;93(7):1167-72. doi: 10.1016/j.apmr.2012.02.026.

Line 391: article citation: op Stroke Rehabil. 2020 May;27(4):251-261. doi: 10.1080/10749357.2019.1691815.

Line 399: citation of article: J Bone Miner Res. 2020 Oct;35(10):1914-1922. doi: 10.1002/jbmr.4048.

Line 400: article citation: Fritz NE, Worstell AM, Kloos AD, Siles AB, White SE, Kegelmeyer DA. Backward walking measures are sensitive to age-related changes in mobility and balance. Gait Posture. 2013 Apr;37(4):593-7. doi: 10.1016/j.gaitpost.2012.09.022.

Line 403: article citation: Phys Ther. 2010 May;90(5):761-73. doi: 10.2522/ptj.20090069.10.1097/NPT.0000000000000210.

Thank you for your feedback. We have revised the citations in our manuscript and adjusted them according to the MDPI ACS style to ensure that all details are complete and the abbreviations as well as authors' names are correctly represented.

Reviewer 3 Report

Comments and Suggestions for Authors

This paper presents an interesting research on the balance assessment of patients with stroke using a 3-meter backword walk test. In general, this paper is well-organized, whereas the following minor revisions are necessary:

1. The authors carefully describe the experimental setup. If a photograph of the experimental setup is provided and labeled, it will be better for the readers to understand and following the current resarch.

2. Most of the experimental results are included in Table 1, e.g., the mean time. The reviewer suggests splitting the results into several tables or tables + figures for a better understanding. For instance, a boxplot or a histogram will be better for the mean +- SD.

3. There are quite a few typo format errors. For instance, line 62 and 63 should be in one line, In Table 1, 11,6 should be 11.6 (same to several numbers)

Author Response

This paper presents an interesting research on the balance assessment of patients with stroke using a 3-meter backword walk test. In general, this paper is well-organized, whereas the following minor revisions are necessary:

  1. The authors carefully describe the experimental setup. If a photograph of the experimental setup is provided and labeled, it will be better for the readers to understand and following the current resarch.

Thank you for your insightful suggestion on enhancing the comprehensibility of our experimental setup by providing a labeled photograph. We agree that a visual representation could significantly aid the readers in understanding our research better. However, unfortunately, we do not have a photograph of the experimental setup. We have tried to describe the setup in detail in the manuscript to provide a clear understanding to the readers.

  1. Most of the experimental results are included in Table 1, e.g., the mean time. The reviewer suggests splitting the results into several tables or tables + figures for a better understanding. For instance, a boxplot or a histogram will be better for the mean +- SD.

Thank you for your suggestion on re-organizing the experimental results. We have opted to split the results into two tables for clearer presentation, while avoiding boxplots or histograms due to space constraints.

  1. There are quite a few typo format errors. For instance, line 62 and 63 should be in one line, In Table 1, 11,6 should be 11.6 (same to several numbers)

Done

Reviewer 4 Report

Comments and Suggestions for Authors

This is a study assessing the measurement properties of the 3MBWT in survivors of stroke. This has been previously studied, nevertheless this study has the potential to add knowledge about the clinimetric properties of the 3MBWT, especially convergent/divergent validity, and robustness of reliability and agreement parameters. I see however, that the terms reliability and validity are at times used interchangeably (e.g., lines 64–65), but are conceptually different. Also, the authors claim that “Furthermore, contrasting the 3MBWT with the 10-Meter Walk Test (10MWT) and the 6 Minute Walk Test (6MWT) is crucial for evaluating its construct validity and establish its distinct place in gait evaluations”; however, it is not clear (i.e., not in depth discussed) if the instruments capture the same aspects of the activity walking (e.g, exercise capacity), or different aspects, or even complementary constructs. Can we use only one? Must we use 2?  Should we use all? Why? What does your research contribute to answer these questions that, partially, are one the reasons for having conducted this research? (Discuss further). If using the same units of measurement, validity can also be assessed using the Bland and Altman method. In fact, it was originally design for comparisons between measuring methods of the same physiological variable [1]

Also, statistics used to assess reliability were not appropriate (e.g., Pearson’s r instead of intraclass correlation coefficients [ICC] as recognized in well renowned authors [2]); and consistency of repeated measurements were conducted using ANOVA tests when agreement statistics are more informative of the degree to which scores or ratings are identical. I suggest that authors follow the directives of GRRAS from Equator network for reporting this type of studies, including the statistical methods recommended there [3] for analyzing interrater/intrarater reliability and agreement studies, re-conduct the analyses, and resubmit the article with improved results section and discussion.

[1] – Bland, J. M., & Altman, D. G. (1986). Statistical methods for assessing agreement between two methods of clinical measurement. Lancet (London, England)1(8476), 307–310.

[2] – Streiner, D.L. &  Kottner, J. (2014)  Recommendations for reporting the results of studies of instrument and scale development and testing. Journal of Advanced Nursing  70(9),  1970–1979. https://doi.org/10.1111/jan.12402

[3] – Kottner, J., Audigé, L., Brorson, S., Donner, A., Gajewski, B. J., Hróbjartsson, A., Roberts, C., Shoukri, M., & Streiner, D. L. (2011). Guidelines for Reporting Reliability and Agreement Studies (GRRAS) were proposed. Journal of clinical epidemiology64(1), 96–106. https://doi.org/10.1016/j.jclinepi.2010.03.002

Author Response

Comments and Suggestions for Authors

This is a study assessing the measurement properties of the 3MBWT in survivors of stroke. This has been previously studied, nevertheless this study has the potential to add knowledge about the clinimetric properties of the 3MBWT, especially convergent/divergent validity, and robustness of reliability and agreement parameters. I see however, that the terms reliability and validity are at times used interchangeably (e.g., lines 64–65), but are conceptually different.

Thank you for your valuable feedback on the distinction between reliability and validity, and for bringing to our attention the misuse of these terms in our manuscript. We have revised the relevant section of our manuscript to ensure clarity and correct usage of these terms. Specifically, we have separated the discussion on reliability from the discussion on validity and introduced a new sentence to transition between these topics. The revised text now reads:

“While its reliability has been validated in earlier studies with stroke survivors [1,2], yet it is essential to continuously confirm this reliability. Additionally, this study aims to examine the concurrent validity of the 3MBWT by comparing it with well-regarded, established assessments like the Functional Gait Assessment (FGA), 10-Meter Walk Test (10MWT), and 6-Minute Walk Test (6MWT) to understand its unique contribution in evaluating gait and balance impairments in stroke survivors. While the 3MBWT has proven reliable post-stroke, its correlation with other known walking function and adaptability outcomes hasn't been explored, which this study intends to address"

Also, the authors claim that “Furthermore, contrasting the 3MBWT with the 10-Meter Walk Test (10MWT) and the 6 Minute Walk Test (6MWT) is crucial for evaluating its construct validity and establish its distinct place in gait evaluations”; however, it is not clear (i.e., not in depth discussed) if the instruments capture the same aspects of the activity walking (e.g, exercise capacity), or different aspects, or even complementary constructs. Can we use only one? Must we use 2?  Should we use all? Why? What does your research contribute to answer these questions that, partially, are one the reasons for having conducted this research? (Discuss further).

Thank you for your invaluable feedback. We have addressed your concerns as detailed below:

  1. Clarification on Gait Aspects Measured by Instruments: We have elaborated in the discussion section on the unique gait aspects captured by the 3MBWT, 10MWT, and 6MWT, underscoring the importance of a multi-faceted gait assessment.
  2. On the Use of One or Multiple Tests: Our manuscript now includes a section explaining the rationale behind selecting three parameters for a thorough evaluation, which provides a clearer understanding of gait impairments in stroke survivors.
  3. Research Contribution: We have highlighted our research's contribution towards better understanding the complementary role these tests play in evaluating gait and balance impairments.

See new paragraph “The assessment of mobility and balance in stroke patients is crucial in the clinical setting for establishing accurate diagnoses, planning individual treatment methods, and evaluating the effectiveness of rehabilitation [28]. The 3MBWT has the potential to provide vital clinically relevant information concerning patients' balance and mobility, aiding in predicting rehabilitation success. In our research, selecting assessments that encapsulate a wide range of gait parameters was crucial to achieve a comprehensive evaluation and to enhance the robustness of our findings. The FGA, 10MWT, and 6MWT were chosen as they spotlight different facets of gait functions, enabling a more nuanced exploration of gait parameters [15,28]. The correlation analysis revealed nuanced insights, underlining the unique contribution of each assessment in evaluating gait and balance impairments in stroke survivors. Our study's results indicated a stronger correlation between the 3MBWT and the 6MWT compared to other tests, despite their different physical constructs. An explanation for this could be that typically stroke survivors exhibit a shortened stance phase and a prolonged swing phase on the paretic side, coupled with a reduced walking speed and stride length [29,30]. These deficits are probably more pronounced in longer tests such as the 6MWT than in shorter tests such as the 10MWT. In addition, longer walking tests may lead to increased neuromuscular fatigue and spasticity [29,31], resulting in a shorter walking distance. Furthermore, studies has shown that walking speed, stride length, and cadence are considerably lower in backward walking compared to forward walking [32]. Unlike forward walking, backward walking requires conscious hip joint extension, which poses a challenge for stroke survivors with hemiplegia, leading to a reduction in the range of motion at the hip joint during backward walking [32]. Moreover, the peak plantar flexion moment at the ankle joint is significantly lower in backward walking, making it difficult to generate propulsion power, which in turn affects walking speed [33]. The observed differences may suggest that tests like the 6MWT, with its longer duration, and the 3MBWT, with its somewhat more complex construct, can detect subtler gait deficits that might not be as easily noticeable in the shorter 10MWT, and therefore might have a marginally stronger correlation.

However, the correlation between the 3MBWT and the FGA was found to be moderate. This suggests that while there is an association, the 3MBWT and the FGA may measure different aspects of gait and balance abilities, indicating that they might not be interchangeable assessments in a clinical setting. The 3MBWT focuses on the ability to walk backward, requiring balance, coordination, and proprioceptive abilities[16]. The FGA, on the other hand, measures a broader spectrum of gait and balance abilities, including the ability to perform head and neck movements while walking. This requires additional skills, such as integrating vestibular information and manoeuvring the body to maintain stability during these added tasks [24]. Consequently, individuals who are capable of efficiently walking backward (and perform well in the 3MBWT) might encounter difficulties in performing complex tasks during the FGA, resulting in a moderate correlation between the two tests.

These differences suggest that backward walking, as measured by the 3MBWT, can reveal more pronounced gait deficits in stroke survivors, thus offering a more nuanced insight into specific gait impairments, especially those provoked by backward walking. Therefore, the 3MBWT could serve as a supplementary tool to provide a more comprehensive evaluation of walking speed, fall risk, and gait impairments in stroke survivors, and potentially be a faster test”.

 If using the same units of measurement, validity can also be assessed using the Bland and Altman method. In fact, it was originally design for comparisons between measuring methods of the same physiological variable [1]

We employed Bland-Altman plots to examine any systematic deviations among the three replicate measurements. Our analysis revealed no significant deviations, so we have included these plots in the Appendix in the manuscript.

Also, statistics used to assess reliability were not appropriate (e.g., Pearson’s r instead of intraclass correlation coefficients [ICC] as recognized in well renowned authors [2]); and consistency of repeated measurements were conducted using ANOVA tests when agreement statistics are more informative of the degree to which scores or ratings are identical. I suggest that authors follow the directives of GRRAS from Equator network for reporting this type of studies, including the statistical methods recommended there [3] for analyzing interrater/intrarater reliability and agreement studies, re-conduct the analyses, and resubmit the article with improved results section and discussion.

Thank you for your continued insightful feedback. Regarding your points on the reliability assessment and the statistical methods employed, we have taken the necessary steps to address these concerns:

See paragraph according ICC “In addition, intra class correlation coefficient (ICC) and its 95%CI was calculated for the three repetitions of 3MBWT.”

Further, the aim of the repeated ANOVA was to detect potential learning or fatigue effects. The corresponding results are now more clearly described in section method as well as discussion. See paragraph in Methods “To investigate the potential for learning or fatigue effects across the three repetitions of the 3MBWT, we conducted an analysis of variance (ANOVA). This allowed us to assess the impact of repeated measures on the test performances.”

As detailed in the methodology section, due to organizational challenges at the rehabilitation center, it wasn't possible to assign the same two physiotherapists for the tests, which precluded the conduct of an inter-rater reliability assessment. However, we aimed to maintain consistency by conducting the tests under identical environmental conditions and applying a "test-retest" design for the 3MBWT.

Reviewer 5 Report

Comments and Suggestions for Authors

In their single-center study "Reliability of the 3-Meter Backward Walk Test and Association 2 with Functional Gait Assessment in Individuals with Stroke" Kapan et al compared  3MBWT, Functional Gait Assessment (FGA), 10-Meter Walk Test (10MWT), and 6-Minute Walk Test (6MWT) between October 2022 and February 2023.

The study is well written and interesting to the reader.

The main conclusion is relevant with the 3MBWT being equivalent to existing walking function tests for stroke patients as well as a time-efficient.

However, I have some minor comments:

1. Please include Abbreviations section.

2. Can the 3MBWT also be used for frail older people for general assasment or is it only used in neurology?

Author Response

The main conclusion is relevant with the 3MBWT being equivalent to existing walking function tests for stroke patients as well as a time-efficient.

However, I have some minor comments:

  1. Please include Abbreviations section.

Done

  1. Can the 3MBWT also be used for frail older people for general assasment or is it only used in neurology?

Thank you for bringing this to our attention. Our study primarily focused on stroke patients and did not extend to the evaluation of frail older individuals. However, the potential application of the 3MBWT in such populations is indeed an intriguing avenue for future research.

Round 2

Reviewer 4 Report

Comments and Suggestions for Authors

The revised version of the manuscript has significantly improved. However, in my point of view it is not suited for publication because of major concerns:

      Authors still present Pearson’s r as a measure of reliability, which is unappropriated (lines 20, 185–186, 227–229 and table 3, 250–251 and 351–352;

      The discussion in lines 271-298 is not in accordance with the authors findings as the magnitude of Pearson’s r is very similar between 3MBWT and 10MWT and 3MBWT and 6MWT;

      I disagree with the authors regarding the interpretation of the Bland and Altman plots. It can be seen in the 3 plots that the greater the mean between two measurements the greater the disagreement, particularly when the mean is above the 20’’. That’s why the limits of agreement are so wide (±5 seconds) considering the mean performance (~11seconds). The consistency is poor because of those extreme outliers that take more than 20 seconds to perform the 3MBWT. Remove these extreme outliers?? Or discuss if the 3MBWT applies to them? What characteristics they have that may help clinicians decide whether to or not to use the 3MBWT in that subgroup??

Other issues (some minor, some major)

Title

Maybe this title is simpler:” Concurrent validity and test-retest reliability of the 3-Meter Backward Walk Test in Stroke Survivors.”

Abstract

Line 20: see my major concern above.

Line 62: replace “validated” by a synonym such as “assessed” or other of your preference, but not to be confused with the measurement property “validadtion”. 

Line 63: “yet it is essential to continuously confirm this reliability” is not a sound justification. Tell us what was missing or less suitable or less controlled or subgroup of patients in previous researches that motivates a new one.

Line 119: Remove “The” before “data”.

Line 172: Put “10 feet” in the metric system.

Lines 187–189: State which model and form of ICC have you used, e.g, ICC2,1 or ICC3,1 or both.

Lines 227–229: Remove these lines including Table 3

Lines 230: Add the model(s) and form(s) of ICC.

Lines 231–234: I disagree with the authors. The greater the mean between the two measurements the greater the disagreement, particularly when the mean is above the 20’’. That’s why the limits of agreement are so wide (±5 seconds) considering the mean performance (~11 seconds). The consistency is poor because of those extreme outliers that take more than 20 seconds to perform the 3MBWT.

Lines 237–240: Help readers interpret the negative and positive Pearson’s r. The longer the time to perform the 3MBWT the longer to perform the 10MWT, the shorter the distance they have covered in the 6MWT and the worse the FGA score. Is that it?

Lines 250–251: see my major concern above.

Lines 271–298: Your results don’t support this rationale as the magnitude of Pearson’s r is very similar between 3MBWT and 10MWT and 3MBWT and 6MWT. Please amend or rebut.

Line 352: see my major concern above.

Line 362: Was this a study objective? It hasn’t been highlighted at the end of introduction. I suggest you renew your conclusions (the all section).

Comments on the Quality of English Language

I provide examples of minor amendments throughout my review.

Author Response

The revised version of the manuscript has significantly improved. However, in my point of view it is not suited for publication because of major concerns:

–      Authors still present Pearson’s r as a measure of reliability, which is unappropriated (lines 20, 185–186, 227–229 and table 3, 250–251 and 351–352;

Thank you for your insightful comments regarding our use of statistical measures in the manuscript. To maintain clarity and avoid confusing our audience with multiple measures of reliability, we have decided to exclude Pearson’s r from our analysis of pairwise comparisons between repeated measurements. Consequently, our discussion focuses solely on the Intraclass Correlation Coefficient as the primary measure of reliability.

See 2.5. Statistical analysis “The Intraclass Correlation Coefficient (ICC) and its 95% Confidence Interval (CI) were calculated for the three repetitions of the 3MBWT using a random effect model, with patient ID as the random effect. The ICC was calculated as the ratio var(random effect) / (var(random effect) + var(residual)). Bland-Altman plots were constructed to examine the limits of agreement and to assess any systematic bias between the sessions [27].”

See Paragraph in Result: “The ICC and 95% CI for the three repetitions were estimated at 0.97 (0.95, 0.98), indicating very small variation within repeated measurements but large variation between patients. The Bland-Altman analysis of the 3-Meter Backward Walk Test (3MBWT) across three attempts highlighted consistent agreement patterns. Moderate agreement was noted between the first and second attempts, with average differences and limits of agreement within acceptable ranges. This trend was similar in comparisons between the first and third, and the second and third attempts. However, an increased spread of differences was observed at higher mean values, particularly above 20 seconds, suggesting the possibility of systematic or proportional errors at longer test durations.”

The discussion in lines 271-298 is not in accordance with the authors findings as the magnitude of Pearson’s r is very similar between 3MBWT and 10MWT and 3MBWT and 6MWT;

Thank you for your valuable feedback regarding the discussion section of our manuscript, particularly concerning the lines 271-298. In light of your comments, we have revisited the discussion section to more accurately reflect the similarity in the correlation coefficients. See revised paragraph.

“The assessment of mobility and balance in stroke patients is crucial in the clinical set-ting for establishing accurate diagnoses, planning individual treatment methods, and evaluating the effectiveness of rehabilitation [29]. In our research, selecting assessments that encapsulate a wide range of gait parameters was crucial to achieve a comprehensive evaluation and to enhance the robustness of our findings. The FGA, 10MWT, and 6MWT were chosen as they spotlight different facets of gait functions, enabling a more nuanced exploration of gait parameters [15,29]. Our results demonstrate similarly strong correlations between the 3MBWT and the 6MWT (r=-0.78), as well as between the 3MBWT and the 10MWT (r=0.71). These similar strengths in correlations suggest that the duration of the 3MBWT is comparably associated with both the distance covered in the 6MWT and the time required in the 10MWT. This may indicate that factors influencing performance in the 3MBWT, such as balance and coordination, are equally relevant for performance in both time-based and distance-based walking tests. While the 6MWT primarily assesses endurance and functional mobility and the 10MWT focuses on walking speed, the similar correlation strengths imply that both aspects – endurance and speed – are closely linked when assessing mobility in stroke survivors. This emphasizes the need to consider both aspects in rehabilitation.

Furthermore, the positive correlation between FGA scores and distances covered in the 6MWT (r = 0.676), along with the negative correlation with times in the 10MWT (r = -0.776), suggests that higher FGA scores correspond to better walking performance. These findings are consistent with the correlations observed between the 3MBWT and the other two tests (6MWT and 10MWT), where longer durations in the 3MBWT indicate reduced walking performance. The similarity of these correlations underscores the complex nature of gait disorders in stroke patients and highlights the need to consider multiple aspects of mobility in clinical assessments.

A strong correlation between the performances in the 6MWT and the 10MWT was anticipated, as previously observed in studies involving stroke patients. These results suggest that both the 6MWT and the 10MWT measure similar aspects of walking ability, largely influenced by neural impairments [25,26]. Additionally the correlation be-tween the 3MBWT and the FGA was found to be moderate. This suggests that while there is an association, the 3MBWT and the FGA may measure different aspects of gait and balance abilities, indicating that they might not be interchangeable assessments in a clinical setting. The 3MBWT focuses on the ability to walk backward, requiring balance, coordination, and proprioceptive abilities [16]. The FGA, on the other hand, measures a broader spectrum of gait and balance abilities, including the ability to per-form head and neck movements while walking. This requires additional skills, such as integrating vestibular information and maneuvering the body to maintain stability during these added tasks [24]. Consequently, individuals who are capable of efficiently walking backward (and perform well in the 3MBWT) might encounter difficulties in performing complex tasks during the FGA, resulting in a moderate correlation between the two tests.

The results suggest that while the various walking tests measure different aspects of gait ability, they are strongly interconnected. This demonstrates the complexity of gait disorders in stroke patients and the necessity to assess multiple parameters in order to obtain a comprehensive picture of mobility.”

I disagree with the authors regarding the interpretation of the Bland and Altman plots. It can be seen in the 3 plots that the greater the mean between two measurements the greater the disagreement, particularly when the mean is above the 20’’. That’s why the limits of agreement are so wide (±5 seconds) considering the mean performance (~11seconds). The consistency is poor because of those extreme outliers that take more than 20 seconds to perform the 3MBWT. Remove these extreme outliers?? Or discuss if the 3MBWT applies to them? What characteristics they have that may help clinicians decide whether to or not to use the 3MBWT in that subgroup??

We agree that the variability of the deviations increase with increasing mean of the two measurements. These points to increased variability of patients, which often exhibit specific characteristics, such as greater post-stroke motor impairment, lower overall physical fitness, or balance issues. These factors contribute to their prolonged test times and, consequently, the increased variability observed in the Bland-Altman plots.

Undoubtedly, the inclusion of these patients with longer test times does expand the variability and, in turn, the width of the limits of agreement. However, we contend that these individuals are not outliers in the conventional sense but rather represent a genuine segment of our study sample. Therefore, we included the few patients thereby avoiding an arbitrary selection of the study sample leading to somehow varnished results with smaller limits of agreement. In the text we point to this no explicitly reading as

“Our analysis of the Bland-Altman plots revealed a range of around ±5 seconds in the limits of agreement, albeit with some outliers. These outliers, possibly associated with patients having more severe post-stroke impairments or advanced age, highlight the natural variability inherent in clinical assessments. Including all patient data in our analysis reflects our commitment to a realistic representation of the 3MBWT's clinical utility, encompassing a broad spectrum of patient profiles and emphasizing the need for nuanced interpretation of test results based on individual patient characteristics."

Other issues (some minor, some major)

Title

Maybe this title is simpler:” Concurrent validity and test-retest reliability of the 3-Meter Backward Walk Test in Stroke Survivors.”

Thank you for your suggestion to revise the title. We have adopted your recommended title, 'Concurrent Validity and Test-Retest Reliability of the 3-Meter Backward Walk Test in Stroke Survivors,' for its clarity and conciseness.

Abstract

Line 20: see my major concern above.

Done “The results indicated that the 3MBWT demonstrated high reliability with an Intra class Correlation Coefficient of 0.97 (95% CI: 0.95, 0.98).”

Line 62: replace “validated” by a synonym such as “assessed” or other of your preference, but not to be confused with the measurement property “validadtion”. 

Done

Line 63: “yet it is essential to continuously confirm this reliability” is not a sound justification. Tell us what was missing or less suitable or less controlled or subgroup of patients in previous researches that motivates a new one.

Thank you for your feedback regarding the justification for re-evaluating the reliability of the 3MBWT. We have revised this section in our manuscript to provide a more detailed rationale.

“While its reliability has been assessed in earlier studies with stroke survivors [20,21], our research seeks to enhance its validity. We aim to achieve this by examining the concurrent validity of the 3MBWT in comparison with other established assessments such as the Functional Gait Assessment (FGA), 10-Meter Walk Test (10MWT), and 6-Minute Walk Test (6MWT). This comparative analysis is crucial to understand the applicability of the 3MBWT in the broader context of stroke rehabilitation, particularly for diverse patient groups with varied gait and balance impairments.

Line 119: Remove “The” before “data”.

Done

Put “10 feet” in the metric system.

Done

Lines 187–189: State which model and form of ICC have you used, e.g, ICC2,1 or ICC3,1 or both.

Following text was added:

ICC was estimated by a random effect model using patient ID as random effect. ICC was calculated as the ratio var(random effect)/( var(random effect)+var(residual)).

Lines 227–229: Remove these lines including Table 3

Done

Lines 230: Add the model(s) and form(s) of ICC.

Please see our comment above.

Lines 231–234: I disagree with the authors. The greater the mean between the two measurements the greater the disagreement, particularly when the mean is above the 20’’. That’s why the limits of agreement are so wide (±5 seconds) considering the mean performance (~11 seconds). The consistency is poor because of those extreme outliers that take more than 20 seconds to perform the 3MBWT.

Please see our response regarding this comment above.

Lines 237–240: Help readers interpret the negative and positive Pearson’s r. The longer the time to perform the 3MBWT the longer to perform the 10MWT, the shorter the distance they have covered in the 6MWT and the worse the FGA score. Is that it?

In the correlation analysis, the 3MBWT showed a negative correlation with the 6MWT (r=-0.78, 95% CI -0.88 to -0.60), indicating that individuals who took longer to complete the 3MBWT tended to cover shorter distances on the 6MWT. A positive correlation was found with the 10MWT (r=0.71, 95% CI 0.49 to 0.84), with longer 3MBWT completion times associated with increased times on the 10MWT. Additionally, a negative correlation with the FGA (r=-0.54, 95% CI -0.74 to -0.26) suggests that longer times on the 3MBWT could be linked to lower FGA scores. Conversely, FGA scores were positively correlated with distances covered in the 6MWT (r = 0.67) and negatively correlated with times in the 10MWT (r = -0.77). These correlations suggest that higher FGA scores are associated with longer distances in the 6MWT and shorter times in the 10MWT. Furthermore, an inverse relationship was identified between the 10MWT times and the 6MWT distances, with a correlation coefficient of -0.86, indicating that longer times in the 10MWT are associated with shorter distances in the 6MWT.

Lines 250–251: see my major concern above.

Done  “In our study, the ICC for the three repetitions of the 3MBWT was estimated at 0.97 (95% CI: 0.95, 0.98), demonstrating minimal variation within repeated measurements, remarkably consistent across five different physiotherapists”.

Lines 271–298: Your results don’t support this rationale as the magnitude of Pearson’s r is very similar between 3MBWT and 10MWT and 3MBWT and 6MWT. Please amend or rebut.

Please see our comment above.

 Line 352: see my major concern above.

See revised text “Although our study showed robust intra-rater reliability, with an ICC of 0.97 (95% CI: 0.95, 0.98) for the three repetitions of the 3MBWT, it is noteworthy that, owing to the study's design, we were unable to directly evaluate inter-rater agreement.

Line 362: Was this a study objective? It hasn’t been highlighted at the end of introduction. I suggest you renew your conclusions (the all section).

See revised text “The study confirms the 3-Meter Backward Walk Test (3MBWT) as a reliable and valid instrument for assessing mobility in stroke survivors, exhibiting high test-retest reliability. It shows a strong concurrent validity with the 10MWT and 6MWT and a moderate correlation with the Functional Gait Assessment (FGA), indicating its effectiveness in capturing distinct yet interrelated aspects of gait and balance. These results endorse the 3MBWT's integration into stroke rehabilitation assessments, providing a comprehensive evaluation of mobility challenges faced by stroke patients, suggesting its potential as a time-efficient alternative”

Round 3

Reviewer 4 Report

Comments and Suggestions for Authors

I'm satisfied with the amendments authors have performed in this version of the manuscript. I think the quality of the reporting has significantly improved.